# Testosterone and Obesity in an Aging Society

**DOI:** 10.3390/biom15111521

**Published:** 2025-10-28

**Authors:** Takahiro Tsutsumi, Kyoichiro Tsuchiya

**Affiliations:** Department of Diabetes and Endocrinology, Graduate School of Interdisciplinary Research, Faculty of Medicine, University of Yamanashi, 1110 Shimokato, Chuo-shi 409-3898, Yamanashi, Japan; tsuchiyak@yamanashi.ac.jp

**Keywords:** obesity, hypogonadism, testosterone, inflammation, aging

## Abstract

Testosterone is a hormone that plays a crucial role in men, maintaining muscle mass and bone density and regulating sexual function. This hormone is associated with the inhibition of obesity and the prevention of obesity-related diseases, such as type 2 diabetes, impaired glucose tolerance, dyslipidemia, hypertension, coronary artery disease, and non-alcoholic fatty liver disease. Obesity has a complex effect on testosterone production and metabolism. Chronic inflammation and hormones associated with obesity cause dysfunction of the hypothalamic-pituitary-gonadal axis, leading to reduced testosterone production. Studies have demonstrated that blood testosterone levels decrease in obese men, suggesting a reciprocal interaction between decreased testosterone and obesity. Additionally, decreased testosterone levels are closely associated with aging. The natural decline in testosterone levels with age can lead to visceral obesity, thus increasing the risk of type 2 diabetes and other chronic metabolic diseases. In many countries, the population is aging, and the importance of testosterone replacement therapy (TRT) for aging men with low testosterone is increasing. Recent studies have expanded our understanding of TRT, highlighting its potential benefits in obese individuals, its interaction with gut microbiota, and the influence of racial differences and genetic polymorphisms on treatment efficacy. This review provides a comprehensive overview of the physiological mechanisms linking obesity and testosterone, current therapeutic approaches including TRT, and emerging research directions that may inform personalized treatment strategies.

## 1. Introduction

Obesity is recognized as a major global public health concern [1,2,3]. Metabolic changes associated with obesity have been shown to increase the risk of chronic diseases, including cardiovascular disease, type 2 diabetes, fatty liver, and sleep apnea [4,5,6,7]. Obesity also affects the endocrine system and is increasingly recognized as a factor contributing to hormonal imbalances [8,9,10,11,12,13,14]. The relationship between obesity and testosterone has been discussed in several clinical and epidemiological studies [15,16].

Testosterone plays essential roles in men, including maintaining muscle mass, bone density, and sexual function. It also helps reduce adiposity and mitigate the risk of related metabolic diseases, including type 2 diabetes, dyslipidemia, and non-alcoholic fatty liver disease [16,17,18,19,20,21,22,23,24,25]. On the one hand, obesity has a complex effect on testosterone production and metabolism [16,17,18]. Specifically, chronic inflammation associated with obesity causes dysfunction of the hypothalamic-pituitary-gonadal (HPG) axis, leading to suppressed testosterone production. Consequently, blood testosterone levels are often reduced in obese men. Thus, there is an interaction between testosterone reduction and obesity [16,17,19].

Testosterone decline is also closely related to aging [15,16,17,18]. Age-related testosterone deficiency, referred to as late-onset hypogonadism, is characterized by low serum testosterone (total or free) and clinical symptoms, including low sexual desire, erectile dysfunction, and reduced capacity for vigorous activity [20]. This condition may also contribute to visceral obesity, further increasing the risk of type 2 diabetes and other chronic metabolic diseases. In this review, we examine the mechanisms underlying age-related testosterone decline. We summarize recent findings on commonly used testosterone replacement therapy (TRT) formulations and evaluate their efficacy and safety, highlighting both potential benefits and concerns, particularly those related to cardiovascular and prostate risks [21,22,23,24,25].

Additionally, emerging evidence indicates that testosterone metabolism may be influenced by the gut microbiota, forming a gut–testis axis, a bidirectional communication pathway between the gut microbiota and testicular function that modulates hormonal balance. Racial differences and genetic polymorphisms may also affect testosterone levels and the efficacy of TRT, highlighting the importance of personalized treatment approaches. This review further discusses recent findings on the roles of gut microbiota and genetic factors in regulating testosterone levels and treatment outcomes. Finally, we present our perspective on future research directions that may contribute to the development of improved clinical strategies.

Overall, this comprehensive review aims to provide insights into the complex relationship between obesity and testosterone, current therapeutic strategies, and emerging topics that may inform individualized care for men with metabolic and hormonal disorders.

For this review, we conducted a literature search using an AI-based search tool in combination with conventional databases, including PubMed and Google Scholar. Keywords included “testosterone,” “obesity,” “hypogonadism,” and “testosterone replacement therapy (TRT),” as well as related terms. The initial search results were screened by the authors, who individually reviewed abstracts and full texts. Inclusion criteria were: (1) human studies or high-quality systematic reviews/meta-analyses, (2) studies published in English, and (3) relevance to testosterone physiology, obesity, or TRT. Exclusion criteria included case reports, non-peer-reviewed articles, conference abstracts, and studies not directly related to the scope of this review. Through this process, inappropriate articles were excluded, resulting in the final set of references.

## 2. Obesity and Testosterone

### 2.1. Obesity Caused by Low Testosterone

#### 2.1.1. Increased Fat Mass

Several studies have investigated the specific mechanisms by which testosterone directly inhibits fat accumulation in adipocytes. Lipoprotein lipase (LPL) promotes fat accumulation in adipocytes by hydrolyzing triglycerides into free fatty acids. Testosterone reduces the LPL activity in adipocytes and regulates fat accumulation, particularly in the visceral fat [26]. In addition, testosterone upregulates beta-adrenergic receptor expression and promotes lipolysis via noradrenaline [27,28]. However, when testosterone levels decrease, this response also decreases, thereby promoting obesity. Additionally, several mechanisms have been reported to suppress fat accumulation through androgen receptors (AR), including the regulation of ELOVL3 [29], Retinol Binding Protein 4 [30], and the modulation of Wnt signaling pathways [31]. Furthermore, indirect effects on adipocytes have been reported. Animal studies suggest that testosterone stimulates physical activity through dopamine pathways and estrogen receptor α (ERα) signaling in the hypothalamus [32,33]. In addition, testosterone differentiates macrophages into the M2 type via the inhibitory regulatory G protein rather than AR. As a result, it has been shown to inhibit the differentiation of preadipocytes [34]. Thus, testosterone inhibits adipose tissue expansion through multiple mechanisms, including reduced adipocyte differentiation and fat formation, as well as enhanced physical activity. Conversely, low testosterone impairs these functions, promoting obesity. Clinical studies consistently demonstrate that low testosterone levels are positively correlated with obesity, particularly with increased visceral fat, which elevates the risk of obesity-related diseases [35,36,37,38,39,40].

#### 2.1.2. Loss of Skeletal Muscle Mass and Decrease in Basal Metabolic Rate

Testosterone plays a pivotal role in maintaining skeletal muscle mass [41,42,43]. It promotes protein synthesis in skeletal muscle [44,45,46], and clinical studies confirm that exogenous testosterone administration exerts anabolic effects on muscle tissue [44,47,48]. In murine models, 5α-dihydrotestosterone (5α-DHT), a potent androgen, has been suggested to act via a putative membrane androgen receptor, GPR133, leading to increased cyclic AMP (cAMP) production and augmented contractile force [49]. Moreover, it stimulates muscle satellite cell proliferation and their differentiation into myoblasts, contributing to muscle regeneration and growth [50,51]. Declines in testosterone impair these processes, leading to muscle atrophy and loss of skeletal muscle mass [52,53]. This reduction in muscle mass lowers the basal metabolic rate, thereby increasing the risk of obesity and related metabolic disorders [54,55,56]. Maintaining adequate testosterone levels is therefore essential for preserving muscle mass, sustaining metabolic rate, and preventing obesity-related complications.

#### 2.1.3. Metabolic Inflammation Caused by Low Testosterone

Obesity is characterized by excessive adipose tissue accumulation, which induces a chronic low-grade inflammatory state known as “metabolic inflammation” [57,58,59]. This condition increases the risk of insulin resistance, type 2 diabetes, cardiovascular diseases, and certain cancers [57,58,59,60,61,62,63,64]. These health problems are particularly pronounced in visceral obesity [59,60,61,62]. Low testosterone contributes to visceral obesity and subsequent metabolic inflammation through several mechanisms. As visceral adiposity progresses, the secretion of pro-inflammatory cytokines (e.g., TNF-α, IL-1β, IL-6) increases [62,63]. Evidence suggests that testosterone deficiency may be associated with elevated inflammatory cytokines (TNF-α, IL-1β, IL-6) in clinical meta-analyses and basic studies [16,17]. TRT has also been shown to reduce the levels of inflammatory cytokines [65,66,67]. These findings suggest that testosterone’s anti-inflammatory effects could contribute to reduced atherosclerosis risk, partly by reducing obesity-related metabolic inflammation [63,64,68,69,70].

During obesity, hypertrophy of adipocytes and alterations in the gut microbiota lead to the excessive production of specific inflammatory ligands, including saturated fatty acids (SFAs), lipopolysaccharides (LPS), and Resistin [71,72,73]. These ligands bind to Toll-like receptor 4 (TLR4) on the surface of cells including macrophages and adipocytes, activating the NF-κB signaling pathway and triggering inflammation [71]. Upon activation of TLR4, inhibitor of κB (IκB) is phosphorylated and ubiquitinated, leading to its degradation by the proteasome. This allows NF-κB to translocate into the nucleus, where it binds to specific DNA sequences and promotes the transcription of pro-inflammatory cytokines. These cytokines, in turn, activate NF-κB through their respective receptors, creating a vicious cycle of inflammation [71,72,73].

Some preclinical studies suggest that testosterone may reduce TLR4 expression and inhibit NF-κB signaling. In orchiectomized mice, TLR4 expression is increased, while testosterone administration reduces TLR4 expression in macrophage cell lines [74,75]. Additionally, testosterone has been reported to inhibit NF-κB signaling in multiple organs [66,76].

Taken together, these findings suggest that testosterone not only indirectly suppresses metabolic inflammation induced by obesity but may also directly regulate inflammatory processes. Elucidating the molecular mechanisms by which testosterone downregulates TLR4 expression and inhibits NF-κB signaling is crucial for developing therapeutic strategies targeting inflammation associated with obesity and metabolic disorders (Figure 1), Appendix A.

#### 2.1.4. The Relationship Between Testosterone and Gut Microbiota

Recent studies have revealed a close association between the gut microbiome and obesity [77,78,79,80,81]. Animal experiments have demonstrated that testosterone influences the composition of the gut microbiota. For example, orchiectomized mice show changes in their gut microbial profile, including an increased Firmicutes/Bacteroidetes ratio and a higher abundance of Lactobacillus, which are commonly associated with obesity [81,82,83]. These microbial shifts have been hypothesized to influence energy absorption and lipid metabolism [82,83].

Although human studies remain limited, systematic reviews have shown that increased testosterone levels are positively correlated with the abundance of Ruminococcus and Acinetobacter species, as well as with greater microbial diversity [84]. Clinical studies suggest that in elderly men, higher testosterone levels are associated with increased abundance of Firmicutes phylum bacteria [85].

A decline in circulating testosterone can alter gut microbiota composition, and conversely, microbiota changes are closely linked to testicular function. This bidirectional relationship, termed the gut–testis axis, is an emerging research focus [77,81,82,86]. Short-chain fatty acids (SCFAs), produced through bacterial fermentation of dietary fiber, may enter the systemic circulation and potentially enhance testosterone synthesis [87,88,89]. One proposed mechanism involves SCFA-induced upregulation of luteinizing hormone receptor (LHR) protein expression and activation of the cAMP/PKA signaling pathway in Leydig cells, which has been shown to stimulate testosterone production [89]. Additionally, certain microbial strains (e.g., Butyricicoccus desmolans, Clostridium scindens) express key steroid-metabolizing enzymes, such as steroid 17,20-desmolase [90,91], and are capable of converting pregnenolone and progesterone into androgenic precursors. These findings suggest that the gut microbiota may function as an extra-gonadal androgen-producing system. On the other hand, the presence of Pseudomonas nitroreducens, a bacterium that degrades testosterone and expresses 3β and 17β hydroxysteroid dehydrogenase (3β and 17β HSD), has been identified. Its potential association with testosterone-induced hyperlipidemia has been suggested [92].

Obesity and the associated metabolic inflammation are associated with gut microbiota alterations [93,94]. This increases intestinal permeability, allowing microbial components such as lipopolysaccharide to enter the bloodstream, which further aggravates metabolic inflammation [95,96]. These inflammatory and metabolic disturbances may also affect the HPG axis and the testes, impairing testosterone production [78,80,97,98,99,100].

Therapeutic strategies that target the gut microbiota, such as probiotics, prebiotics, dietary interventions, and fecal microbiota transplantation, have shown promise in improving testosterone levels and metabolic health in men with functional hypogonadism by ameliorating inflammation caused by harmful gut microbiota [78,80,81,82]. However, the current quality of human evidence remains limited. Most clinical data are observational or derived from studies with small sample sizes and heterogeneous populations. While the effects of TRT on the gut microbiome have been explored in transgender individuals [101], robust interventional trials in hypogonadal men are lacking. A deeper understanding of the gut–testis axis may lead to personalized treatments for metabolic syndrome, type 2 diabetes, and age-related androgen decline. To achieve this, future research must address translational challenges, including the variability in microbiota composition across individuals, the complexity of host–microbe interactions, and the need for standardized protocols.

### 2.2. Testosterone Decline Caused by Obesity

#### 2.2.1. Leptin Resistance

Leptin is a peptide hormone secreted primarily by adipocytes. It acts on hypothalamic kisspeptin neurons to stimulate gonadotropin-releasing hormone (GnRH) secretion [102,103,104,105,106,107,108]. Specifically, leptin binds to receptors (LEPR) on kisspeptin neurons, enhancing kisspeptin release. However, in obesity, leptin resistance develops, LEPR activity decreases, and kisspeptin secretion declines. Consequently, GnRH neuron signaling via the KiSS-1 receptor (Kiss1R) diminishes, reducing GnRH output [103,104,106]. Additionally, some studies have shown that high levels of leptin reduce testosterone production in the testis [109]. LEPRs are expressed in Leydig cells, and elevated leptin reduces expression of steroidogenic acute regulatory protein (StAR), impairing testosterone synthesis [109]. Thus, obesity-induced leptin resistance and hyperleptinemia decrease testosterone production through central and peripheral mechanisms.

#### 2.2.2. Metabolic Inflammation Associated with Obesity

Obesity is associated with low-grade inflammation characterized by elevated inflammatory cytokine production, particularly from visceral adipose tissue [7,93]. Overproduction of cytokines (e.g., TNF-α, IL-1β, IL-6) not only contributes to metabolic complications but is also linked to reduced testosterone levels [93,97,110,111].

Pro-inflammatory cytokines impair the HPG axis. Long-term TNF-α exposure suppresses Kiss1R gene expression, disrupts kisspeptin signaling, and reduces GnRH secretion in hypothalamic cells [99]. Similarly, IL-1β interferes with GnRH mRNA translation and inhibits luteinizing hormone (LH) release [100]. In addition, cytokines directly impair Leydig cell steroidogenesis: cultured Leydig cells exposed to TNF-α, IL-1β, or IL-6 produce less testosterone [110,112]. Obesity also increases reactive oxygen species (ROS) production [113]. ROS induce mitochondrial dysfunction and oxidative stress in Leydig cells, inhibiting StAR activity [114,115,116]. Reduced StAR function decreases cholesterol-to-testosterone conversion, further lowering testosterone levels.

#### 2.2.3. Increased Estradiol Production

The relationship between obesity, estradiol, and hypothalamic function in men is complex. Early studies reported higher circulating estradiol levels in obese men compared with lean controls [117], attributed to increased aromatization of testosterone to estradiol in adipose tissue. Elevated estradiol then suppresses GnRH secretion via negative feedback on the HPG axis, reducing LH and follicle-stimulating hormone (FSH) secretion and thereby decreasing testicular testosterone production [16]. However, subsequent studies have challenged the universality of this mechanism. Some reports indicate that estradiol levels are not consistently elevated in obese hypogonadal men and may even be lower than in non-obese men [118]. Moreover, weight loss interventions have not always resulted in significant reductions in plasma estradiol levels [119]. Elevated estradiol levels may also contribute to HPG suppression in some obese men, as some guidelines mention the use of aromatase inhibitors as a potential option in specific cases [120]. However, leptin resistance and inflammation are likely to represent more dominant and consistent mechanisms [17].

#### 2.2.4. Reduction in Sex Hormone-Binding Globulin (SHBG)

SHBG is a glycoprotein that binds and transports sex hormones, including testosterone and estrogen, in the circulation [121]. Obesity is associated with marked reductions in SHBG levels, which directly affect testosterone bioavailability [122,123]. Specifically, SHBG decreases by approximately 1.26 nmol/L for every one-unit increase in body mass index (BMI) [124]. Obese men exhibit significantly lower serum testosterone and SHBG compared with non-obese men, suggesting that obesity exerts direct effects on androgen status [125]. SHBG is synthesized in the liver, and its production is reduced by factors linked to obesity, including insulin resistance and inflammatory cytokines (TNF-α, IL-1β, IL-6) [121]. Therefore, decreased SHBG is considered one of the major contributors to obesity-associated hypogonadism [15,16,17] (Figure 2).

## 3. Aging and Testosterone

### 3.1. Age-Related Testosterone Decline and Its Mechanisms

#### 3.1.1. Testicular Dysfunction

Leydig cells, which produce testosterone in the testes, gradually lose function with aging [126,127,128,129]. Testosterone synthesis in Leydig cells involves LH, StAR, translocation protein (TSPO), and a series of steroid hormone synthases (CYP11A1, 3β-HSD, CYP17A1, and 17β-HSD). When LH acts on Leydig cells, it activates the cAMP pathway. This activation promotes the function of StAR protein and transports cholesterol into the mitochondrial membrane. TSPO is also involved in cholesterol transport to the mitochondrial membrane. Within mitochondria, cholesterol is progressively converted by various steroid hormone synthase to produce pregnenolone and eventually testosterone. In aging men, a reduced responsiveness to LH stimulation has been reported to impair the steroidogenic pathway in Leydig cells [129,130,131]. Furthermore, decreased cAMP production [130,131,132], reduced StAR expression [133,134], lower TSPO levels, and diminished activity of steroidogenic enzymes [135,136,137,138,139] have all been observed. In addition, studies have demonstrated a decline in the number of Leydig cells in older men [140], further contributing to reduced testosterone production capacity.

#### 3.1.2. Dysfunction of the HPG Axis

GnRH secreted by the hypothalamus stimulates LH secretion from the pituitary gland, which in turn promotes testosterone production in the testes. In aging men, GnRH is secreted in an irregular pattern with reduced mean amplitude, resulting in lower overall secretion [15,141]. Consequently, LH secretion becomes irregular and decreases. This reduction in LH secretion is attributed to age-related changes in the hypothalamus and pituitary gland, particularly a decline in the number and function of GnRH neurons and degeneration of kisspeptin neurons. These combined factors reduce LH secretion [15]. However, pituitary responsiveness to GnRH is not affected by age, suggesting that hypothalamic changes are the primary cause [142,143]. These mechanisms are further complicated by the involvement of obesity and comorbidities in addition to the effects of aging (Figure 3).

### 3.2. “Primary Hypogonadism” and “Secondary Hypogonadism”

Low testosterone with aging is classified into “Primary hypogonadism,” mainly caused by testicular dysfunction, and “Secondary hypogonadism,” caused by hypothalamic dysfunction [15]. Primary hypogonadism refers to a condition in which blood testosterone levels are low and serum LH levels are elevated simultaneously [144,145,146]. This is primarily due to age-related testicular dysfunction. The increase in LH levels in this pathology is thought to be due to a decrease in negative feedback to the HPG axis by testosterone and estradiol, which is converted from testosterone by aromatase [15]. AR and estrogen receptor-α (ERα) are present in kisspeptin neurons in the hypothalamus [15,147]. Some studies have reported absent AR-mediated negative feedback of testosterone at the pituitary level [148,149]. Collectively, these findings suggest that testosterone and estrogen suppress kisspeptin neurons in the hypothalamus, thereby exerting negative feedback on the HPG axis. Some studies have shown increased kisspeptin signaling in the hypothalamus of aging men, possibly due to reduced negative feedback from testosterone [150,151]. Therefore, it is thought that low testosterone levels due to decreased testicular function reduce this feedback mechanism and increase the blood LH levels. Primary hypogonadism has been observed to increase in men aged 65–70 years and older, and TRT for them may exert similar effects on primary hypogonadism in younger men [15].

In contrast, secondary hypogonadism refers to a condition in which serum testosterone levels are low, but serum LH levels are not elevated [15]. Its primary pathology lies in dysfunction of the HPG axis, and it is frequently associated with obesity [152]. In such cases, clinicians should prioritize lifestyle interventions targeting weight reduction and overall health improvement [153]. It is also necessary to consider the possibility that age-related decline in hypothalamic function may be the cause. Changes in LH levels in aging men vary depending on the individual case. If the LH level is low in low testosterone aging men, secondary hypogonadism is suspected. If it is high, primary hypogonadism is suspected. However, it is important to note that it is difficult to distinguish between these states clearly. As all aging men have age-related changes in the testis and hypothalamus, this classification only indicates which factor is more pronounced. Therefore, aging men should be considered to have both primary and secondary hypogonadism [154,155,156]. Careful clinical assessment and comprehensive diagnosis are required.

### 3.3. Differences Between TRT for Obesity and TRT for Age-Related Testosterone Decline

Currently, evidence on the efficacy and safety of TRT in obese men with low testosterone is insufficient [157,158,159]. Many studies have demonstrated that achieving weight reduction to an appropriate level can improve endogenous testosterone concentrations. For example, weight loss induced by calorie restriction has been shown to significantly increase testosterone levels in obese men [160,161,162]. Similarly, aerobic exercise and resistance training may contribute to the restoration of endogenous testosterone secretion by reducing body fat [163,164]. Accordingly, current guidelines do not recommend TRT as a treatment for obesity per se, and its use requires careful consideration [158,165,166]. However, some recent studies have advocated differently. TRT has been reported to reduce obesity through fat loss and skeletal muscle mass gain, and to have beneficial effects on the reduction of cardiovascular risk factors [165,166,167]. Thus, some obese men may benefit from TRT as part of a broader health strategy. Further well-designed randomized controlled trials and mechanistic studies are warranted to clarify the risks and benefits of TRT in obese men. Until such data are available, healthcare professionals should reserve TRT for men with confirmed hypogonadism (symptoms plus consistently low testosterone levels), ensure ongoing monitoring, and strongly encourage lifestyle modifications as the primary approach to weight management.

In contrast, the approach differs for age-related testosterone decline. After approximately age 65–70, Leydig cell function progressively decreases, leading to a higher prevalence of primary hypogonadism, characterized by low serum testosterone and elevated LH. As age-related primary hypogonadism is typically irreversible, TRT may be an effective treatment [22,168]. Similarly, even in secondary hypogonadism without elevated LH, if hypothalamic decline with aging is irreversible, TRT may be required. However, in some aging men, hypothalamic and/or testis functions may be preserved. Improving obesity may improve low testosterone status in these men. Thus, in obese older men with low testosterone, lifestyle intervention should be the first-line strategy.

The indication for TRT in older men requires careful consideration. TRT should be considered for those with irreversible primary hypogonadism (low testosterone and elevated LH) who are symptomatic. Even in secondary hypogonadism without elevated LH, TRT may be appropriate if due to an irreversible hypothalamic decline. Before starting TRT, clinicians must rule out reversible causes such as systemic illness, medication, or pituitary disorders. In addition, obese men should not be completely excluded from TRT indications. It is also recommended to consider TRT for obese men if the benefits for improving symptoms are greater than the risk of treatment, while managing obesity at the same time. However, although some studies suggest potential benefits of TRT in obese men [165,166,167,168], the overall evidence remains insufficient to draw firm conclusions regarding efficacy and safety.

In conclusion, treatment decisions should be individualized and focused on symptom improvement. Additionally, in obese men, it is desirable to encourage lifestyle improvements and implement interventions. In addition, it is recommended to consider TRT when the benefits outweigh the risks.

## 4. Testosterone Replacement Therapy (TRT)

### 4.1. Current Formulations for TRT

A variety of TRT formulations are currently available [157,159,169]. Each formulation has its unique advantages and disadvantages, and the right choice must be made based on the patient’s lifestyle, treatment goals, risk of side effects, and economic factors.

Intranasal gels are noninvasive, easy to use, and allow the maintenance of a relatively stable level of testosterone in the blood [170,171,172]. They also avoid the first-pass effect in the liver. However, frequent doses are required, and this may affect patient adherence. In addition, local side effects, such as irritation of the nasal mucosa, have been reported.

Transdermal testosterone gels are self-administered, flexible in dosing, and generally well accepted by patients [173,174,175]. However, there is a risk of secondary exposure by touching another person with the hand where the drug was applied, and there is a possibility of skin irritation. It has also been suggested that the blood testosterone levels may fluctuate [157,169]. Transdermal testosterone ointment, like a gel, is a noninvasive and convenient method of administration [176,177]. The texture is easy to apply and blend into the skin, making it suitable for individualized treatment. However, as with gels, the risk of secondary exposure to others, rough skin, and blood testosterone levels may fluctuate.

Oral testosterone tablets are convenient to carry and store [178]. However, they undergo first-pass hepatic metabolism, leading to fluctuations in serum levels and potential systemic side effects such as hepatotoxicity, polycythemia, hypertension, and acne [157,179].

Intramuscular injections are relatively inexpensive, long-acting, and maintain stable serum testosterone levels [157,179,180]. In addition, because frequent dosing is not required, it can be expected that patients’ medication adherence will improve [180]. However, they are associated with pain at the injection site, injection-site reactions, and systemic side effects, including polycythemia, hypertension, and acne, as well as potential impairment of fertility [179,181].

Subcutaneous injections are a relatively new option [180,182,183]. It has been suggested that this treatment has a stable and predictable concentration of the drug during treatment. Additionally, it is better accepted by patients, easier to self-administer than intramuscular injection. In addition, local side effects that occur at the injection site are mild and most likely transient. However, they require patients to master self-injection techniques, which may reduce adherence [157,179]. Long-term studies with larger populations are still needed to evaluate their safety and efficacy.

Testosterone pellets provide prolonged testosterone release, requiring infrequent administration and improving adherence [184,185,186,187]. However, implantation is invasive, carries a risk of infection or local inflammation, and may negatively impact fertility [179].

To choose the best treatment among these options, a thorough consultation between the clinician and patient is essential (Table 1).

### 4.2. Short-Acting and Long-Acting TRT Drugs

TRT drugs may be classified into short -acting and long-acting drugs, depending on the difference in the action time of the formulation and the characteristics of the side effects [188,189,190]. Short- and long-acting testosterone drugs have different characteristics and are considered important factors in the choice of treatment [190].

Short-acting formulations require multiple daily doses but mimic physiological testosterone secretion [172,188]. Therefore, it has a low impact on male fertility and minimizes the suppression of spermatogenesis and LH [188]. This property may allow for the treatment of hypogonadism while maintaining fertility. They may also reduce the risk of secondary erythrocytosis [171]. However, owing to the need for frequent dosing, patient adherence can be challenging. Medications include intranasal gels, transdermal testosterone gels, transdermal testosterone ointments, and oral testosterone tablets [189,190].

On the other hand, Long-acting formulations require less frequent dosing, improve adherence, and maintain more stable testosterone levels [189,190]. However, this formulation may suppress spermatogenesis by suppressing the HPG axis, thereby increasing the risk of infertility [188,191,192]. In addition, the risk of polycythemia has been noted [157,179]. Medications include intramuscular injections, subcutaneous injections, and testosterone pellets.

Clinicians should balance efficacy, safety, and patient preference when selecting an appropriate formulation.

### 4.3. Recent Perspectives on the Side Effects of TRT

#### 4.3.1. Cardiovascular Disease

In the past, there have been concerns that TRT increases the risk of cardiovascular disease, but recent large-scale studies are challenging the previously assumed link [25,193,194]. There are many reports that TRT within the physiological range does not increase the risk of cardiovascular events, and its safety is supported [25,194]. The Testosterone Replacement Therapy for Assessment of Long-term Vascular Events and Efficacy Response in Hypogonadal Men (TRAVERSE) study evaluated the cardiovascular safety of TRT in men with hypogonadism or high-risk and cardiovascular disease [25]. The results showed that TRT was non-inferior to placebo in the incidence of major adverse cardiac events. On the other hand, the study reported an increased incidence of non-fatal arrhythmias requiring intervention, including atrial fibrillation. However, meta-analyses in several other studies have shown that the risk of atrial fibrillation with TRT was non-inferior compared to placebo [194]. This result suggests that there is a poor association between TRT and an increased risk of atrial fibrillation. In addition, TRT for type 2 diabetes mellitus and metabolic syndrome men has been shown to reduce triglycerides (TG) and low-density lipoprotein (LDL) and improve high-density lipoprotein (HDL) and may reduce cardiovascular risk [195,196,197,198,199].

From these recent studies, the epidemiological evidence supporting that TRT does not increase cardiovascular risk is strong. These findings are likely attributable to improvements in obesity associated with TRT, and TRT in obese patients may exert a protective effect against cardiovascular disease risk. However, lifestyle-based weight reduction remains the first-line intervention for obese men with low testosterone, and TRT should be considered when hypogonadal symptoms persist despite such measures. For patients outside of these categories, such as non-obese older men, a different interpretation is required. Moreover, evidence regarding patients with underlying medical conditions is insufficient and warrants further investigation. Caution is particularly needed in non-obese older men with multiple comorbidities. A recent report has shown that the prevalence of erectile dysfunction (ED) in men with coronary artery disease reaches 90%, which may increase the demand for TRT [200]. Further large-scale studies are required to verify the safety and effectiveness of TRT in such patients.

#### 4.3.2. Prostate Disease

The association between TRT and prostate cancer has been extensively debated. Recent evidence suggests that TRT does not increase the risk of prostate cancer [201,202,203]. A randomized controlled trial in middle-aged and older hypogonadal men found no significant difference in high-grade prostate cancer or other prostate-related events between TRT and placebo [203]. However, careful judgment is required when applying TRT in patients with a history of prostate cancer [201,204]. Specifically, TRT is not recommended for the presence of unassessed prostate nodules or induration, or for PSA levels greater than 4 ng/mL (greater than 3 ng/mL for African Americans and men with a family history of high-risk prostate cancer).

Furthermore, recent studies have shown that TRT does not significantly increase the risk of benign prostatic hyperplasia (BPH), especially in men with hypogonadism. For example, a study by Bhasin et al. and a study by Rastrelli et al. suggested that TRT may not cause worsening of lower urinary tract symptoms (LUTS) or an increase in prostate size in men who have been adequately screened for prostate cancer risk [201,205]. However, some studies suggest that testosterone may contribute to prostate growth, and individual responses to TRT can vary greatly [206,207]. These findings show a complex relationship between testosterone levels and prostate health.

Although numerous studies have investigated the cardiovascular and prostate effects of TRT, the evidence remains heterogeneous [25,193,202,203,208]. Clinical trials and observational studies differ substantially in study design, sample size, population characteristics, and follow-up duration. For example, while some randomized controlled trials have suggested neutral or even beneficial effects of TRT on cardiovascular outcomes, others have reported potential risks, particularly in older men with preexisting cardiovascular disease [193,209]. Similarly, findings regarding prostate safety are inconsistent, partly due to variations in baseline risk profiles, inclusion criteria, and outcome definitions. These discrepancies highlight the importance of recognizing methodological limitations, potential biases, and population heterogeneity when interpreting the literature. Further large-scale, long-term trials with standardized endpoints are warranted to establish a clearer understanding of TRT safety in these areas of controversy.

#### 4.3.3. Other Adverse Effects

TRT has traditionally been considered beneficial for improving bone mineral density. However, a recent sub-study of the TRAVERSE trial, which included 5204 participants, reported an unexpected increase in the incidence of clinical fractures, particularly traumatic fractures involving the ribs, wrists, and ankles [210]. This finding challenges previous assumptions and warrants closer examination. Importantly, the TRAVERSE trial was not originally designed to investigate fracture risk or elucidate the underlying biological mechanisms. As such, the causal relationship between TRT and increased fracture risk remains unclear. Several speculative mechanisms have been proposed. One possibility is that TRT-induced improvements in mood and energy levels may lead to increased physical activity [211,212], inadvertently raising the risk of trauma-related fractures. Another consideration is the effect of testosterone on bone turnover. Changes in the balance between bone formation and resorption could influence bone strength in ways that are not yet fully understood [213,214]. It is also possible that this observation was coincidental and not directly related to TRT [215]. Further research is needed to clarify these possibilities.

Beyond fracture risk, other adverse effects of TRT have been reported, including polycythemia, sleep apnea, fluid retention, peripheral edema, and, in some cases, heart failure [157,201,216]. Due to individual variability, TRT should be tailored to each patient’s comorbidities and genetic profile. Personalized approaches may help minimize risks while maximizing therapeutic benefits.

### 4.4. Comparison of the Guidelines for Testosterone Replacement Therapy (TRT)

Currently, official guidelines for TRT are provided by organizations: the Endocrine Society [179], the American Urological Association (AUA) [120], the International Society for the Study of the Aging Male (ISSAM) [217], and the European Menopause and Andropause Society (EMAS) [159]. These guidelines are periodically updated. In this section, we summarize their commonalities, differences, and distinctive features based on the latest information (Table 2).

A common issue among all guidelines is that TRT can suppress spermatogenesis and reduce fertility. Therefore, young men with hypogonadism should receive counseling regarding potential fertility concerns prior to treatment. In addition, regular monitoring of serum testosterone levels, hematocrit (Ht), and prostate-specific antigen (PSA) is recommended both before and after initiating TRT.

A major difference among the guidelines lies in the diagnostic cut-off criteria for total testosterone levels. The Endocrine Society emphasizes the presence of compatible clinical symptoms together with consistently low morning total testosterone levels confirmed on repeated testing, rather than defining a strict numerical threshold. Harmonized reference ranges from population studies suggest a lower limit of approximately 264 ng/dL (9.2 nmol/L) in healthy young men. The AUA recommends a total testosterone level <300 ng/dL (10.4 nmol/L) as a reasonable cut-off to support the diagnosis of hypogonadism. ISSAM suggests considering TRT when morning values (between 7 a.m. and 11 a.m.) are below approximately 350 ng/dL (12.1 nmol/L). EMAS considers values <8 nmol/L (≈231 ng/dL) as indicative of a high probability of hypogonadism, whereas values between 8–12 nmol/L (≈231–345 ng/dL) are considered borderline, warranting further assessment of free testosterone levels. Regarding free testosterone, the Endocrine Society recommends measurement using equilibrium dialysis or calculated values based on total testosterone, SHBG, and albumin concentrations. ISSAM refers to a lower limit of approximately 220–347 pmol/L (≈63.5–100 pg/mL). EMAS considers a free testosterone level <225 pmol/L (≈65 pg/mL) as supportive evidence of hypogonadism when total testosterone is borderline. Because reference values differ depending on assay and calculation methods, results should be interpreted using standardized methods and consistent units.

In their characteristic statements, the Endocrine Society highlights the potential role of TRT in specific secondary hypogonadal conditions, including men with HIV-associated hypogonadism and those with opioid-induced hypogonadism. The AUA emphasizes the importance of counseling both before and during TRT, particularly in men concerned with fertility, and suggests alternative treatment options such as selective estrogen receptor modulators (SERMs), human chorionic gonadotropin (hCG), and aromatase inhibitors. EMAS places special emphasis on bone health, addressing osteoporosis and fracture risk, and explicitly incorporates bone density assessment into its recommended monitoring schedules. ISSAM particularly discusses the use of long-acting injectable testosterone undecanoate and transdermal testosterone gel, while expressing a negative view of oral testosterone formulations due to safety and pharmacokinetic concerns. ISSAM also notes that androgen receptor CAG repeat polymorphisms may modulate individual responses to TRT. As outlined above, each guideline presents both similarities and distinctive perspectives, providing recommendations on TRT from its respective viewpoint.

### 4.5. Racial Differences in Response to TRT

Racial and ethnic backgrounds may influence baseline testosterone levels and clinical responses to TRT. Several studies have reported significant interethnic differences in circulating testosterone. African men and women generally have higher testosterone levels than their White counterparts, even after adjusting for age and BMI [218]. Hispanic men also show elevated levels compared to White men, though the difference is less pronounced [218]. In comparisons between American and Chinese men, total testosterone is higher in Americans, while free and bioavailable testosterone levels show no significant differences [219]. Lifestyle and cultural factors such as diet, physical activity, alcohol intake, and sleep patterns also vary across populations and may interact with endocrine and metabolic pathways relevant to obesity and testosterone regulation [220,221,222,223,224,225].

SHBG concentrations, which affect testosterone bioavailability, differ by race and may influence free testosterone levels and TRT outcomes. For example, White men have lower age-adjusted SHBG levels than African men [226]. The CAG repeat length in the AR gene is a determinant of androgen sensitivity. Shorter repeats are associated with higher receptor activity [227,228,229,230]. African American men typically have shorter repeats (18–20), suggesting greater AR activity, while Caucasian men average 21–22 repeats. East Asian men tend to have longer repeats (23–24), which may indicate reduced AR activity. Hispanic men show similar lengths to Caucasians, around 21 [231,232]. These genetic differences may contribute to variable responses to TRT across populations.

Despite these findings, there is a paucity of comparative studies directly assessing the efficacy and safety of TRT across diverse racial and ethnic populations. Large-scale, multi-ethnic cohort studies and randomized controlled trials are needed to elucidate how racial and ethnic backgrounds influence the relationship between testosterone and obesity, thereby informing more personalized treatment strategies.

## 5. Discussion and Future Research Topics

### 5.1. Elucidation of the Causal Relationship Between Obesity and Low Testosterone

A detailed genetic-level analysis is required to better understand the relationship between obesity and low testosterone. Previous studies have shown that testosterone inhibits fat accumulation and promotes lipolysis in adipocytes [26,28,31,34]. However, the direct roles of AR target genes remain poorly understood. In addition, the transcriptional activity of AR is strongly influenced by transcription factors [233,234], and research on transcriptional factors involved in AR action in adipocytes has been insufficient. Recent studies have also indicated that AR functions not only as a nuclear receptor but also as a membrane receptor, and the role of membrane AR in adipocytes warrants further investigation [49]. In addition, mouse models with adipocyte-specific AR knockout (ARKO) mice have been provided at the individual level; however, adipocyte ARKO mouse models did not show an increase in fat mass [235]. The reason for this is thought to be the influence of organs other than adipocytes, such as the immune effect of blood cells derived from the bone marrow [236,237]. Therefore, a proper understanding of AR function in adipocytes and obesity requires careful distinction between cellular- and organism-level mechanisms. Moreover, CAG repeat polymorphisms in the AR gene have been shown to correlate with reduced AR transcriptional activity [227,238,239]. The impact of CAG repeat number on AR activity in adipocytes, and its association with obesity and visceral fat accumulation in humans, is not yet fully understood. To clarify this, further studies using animal and cell models are necessary.

### 5.2. Search for TRT Protocols That Maintain Reproductive Function

Fertility is an important component of men’s quality of life, especially in countries with declining birthrates. Reduced reproductive function is a major concern with TRT, as men with low testosterone must choose between the benefits of TRT (e.g., improved mental health) and the preservation of fertility. Thus, there is a need to develop TRT strategies that minimize adverse effects on fertility. Short-acting TRT formulations may help preserve fertility [172,176,188,190]. Intranasal gels, although supported by less evidence than other formulations, have been proposed as a newer option that may preserve reproductive function [189]. In addition, certain agents traditionally used to treat azoospermia have been considered for use alongside TRT. Specifically, aromatase inhibitors, combination therapy with human chorionic gonadotropin (hCG) and testosterone, and selective estrogen receptor modulators (SERMs) may be effective [240,241]. These approaches may help improve patient quality of life while preserving future fertility potential.

### 5.3. The Need for Revalidation of Guidelines for TRT in Aging Men

An unexpected finding of the TRAVERSE study was an increased risk of fractures, despite previous evidence that TRT improves bone density [210]. One possibility is that the mood-enhancing effects of TRT increased activity levels, but the causal relationship between TRT and fracture risk requires further investigation. Thus, TRT in aging men may involve unanticipated adverse effects, particularly those influenced by age-specific factors such as comorbidities and concurrent medications. To promptly detect and address such risks, careful and extensive data collection is required. In addition, the current TRT uses only testosterone concentration (blood testosterone level) as an indicator; however, this may not be sufficient. Age is associated with higher SHBG levels, and total testosterone binding to SHBG in blood levels is insufficient to assess biologically active testosterone in the elderly [242,243,244]. Free testosterone levels, which reflect biologically active testosterone rather than total testosterone levels, have been used; however, the credibility of clinically convenient immunoassay methods is questionable [245,246]. Furthermore, some patients experience persistent symptoms despite reaching target testosterone levels during replacement therapy [247,248]. These issues suggest the need for new evaluation methods that assess not only testosterone levels but also biological effects.

Emerging digital health technologies offer promising solutions for monitoring TRT efficacy and safety. Wearable devices can track physical activity, sleep patterns, heart rate variability, and other physiological parameters in real time, providing valuable insights into treatment responses and potential adverse effects [249]. Additionally, advanced biomarkers such as metabolomic and proteomic profiles may help assess tissue-level androgen activity and identify early signs of treatment-related complications [250,251]. Integrating these tools into clinical practice could enable more personalized and adaptive TRT monitoring and strategies, particularly for aging populations with complex health needs.

Addressing these challenges will support the development of safer and more effective TRT protocols. Such progress may contribute to improved management of obesity and age-related hormonal imbalances. Future advancements will depend on translational research and well-designed randomized clinical trials with clearly defined endpoints.

## 6. Conclusions

Obesity and testosterone deficiency are major challenges in modern medicine, and elucidating their interaction may lead to the development of novel therapeutic strategies. In aging societies, addressing these issues may contribute to maintaining both individual and public health. To deepen our understanding in this field, continued research and discussion are essential. Future studies should prioritize long-term randomized controlled trials of TRT in obese men, interventional approaches targeting the gut microbiome, and the integration of genetic and metabolomic data into personalized clinical practice, which may pave the way for more effective and individualized treatments.

## Figures and Tables

**Figure 1 biomolecules-15-01521-f001:**
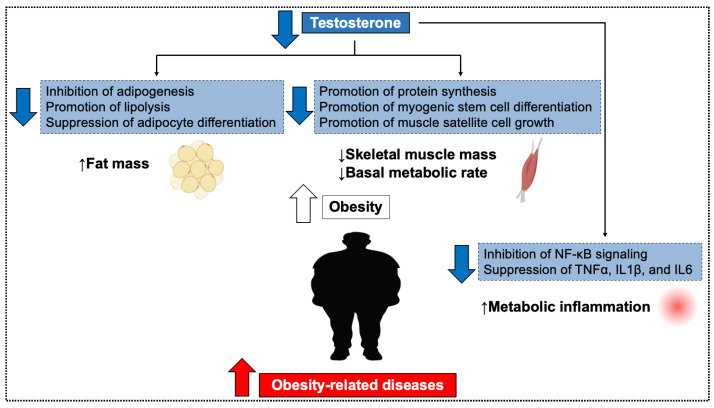
Obesity caused by low testosterone. Decreased testosterone action results in increased fat mass, loss of skeletal muscle mass, and a decrease in basal metabolic rate. Additionally, it increases metabolic inflammation. As a result, the risk of obesity-related diseases is elevated.

**Figure 2 biomolecules-15-01521-f002:**
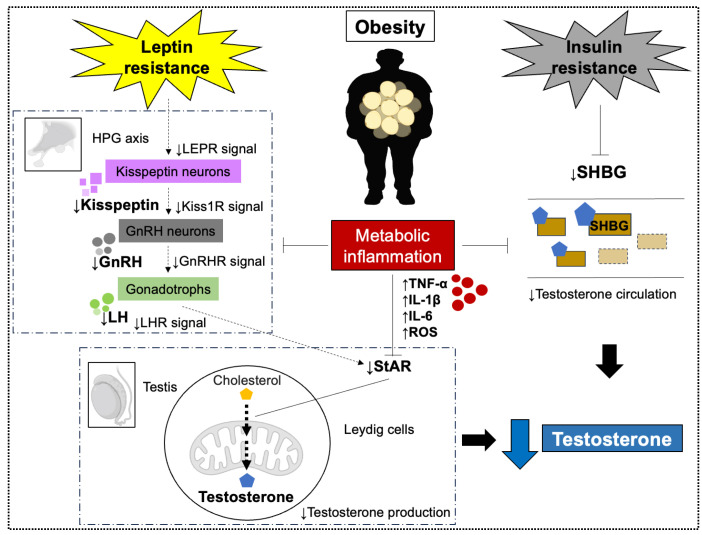
Testosterone decline caused by obesity. Obesity-induced increases in Leptin resistance and insulin resistance, metabolic inflammation, inhibit various mechanisms related to testosterone synthesis in the hypothalamic-pituitary-gonadal (HPG) axis and testes. As a result, the amount of testosterone produced is decreased. Abbreviations: Leptin receptor (LEPR); KiSS-1 receptor (Kiss1R); Gonadotropin-releasing hormone (GnRH); GnRH receptor (GnRHR); Luteinizing hormone (LH); LH receptor (LHR); reactive oxygen species (ROS); steroidogenic acute regulatory protein (StAR); sex hormone-binding globulin (SHBG).

**Figure 3 biomolecules-15-01521-f003:**
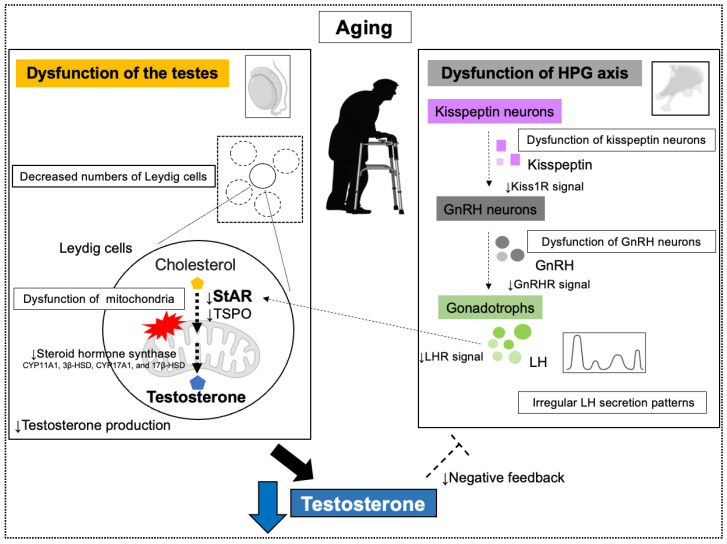
Age-induced testosterone decline and its mechanism. Age-related dysfunction of the hypothalamic-pituitary-gonadal (HPG) axis and dysfunction of the testes reduce testosterone synthesis. Abbreviations: steroidogenic acute regulatory protein (StAR); translocation protein (TSPO); Kisspeptin Receptor (Kiss 1R); Gonadotropin-releasing hormone (GnRH); Luteinizing hormone (LH); LH receptor (LHR).

**Table 1 biomolecules-15-01521-t001:** The Testosterone Replacement Therapy (TRT) formulations currently in use and their advantages and disadvantages.

The TRT Formulations	Advantages	Disadvantages
Intranasal gels	- Maintain stable blood testosterone levels - Avoids first-pass	- Multiple administrations - Local side effects such as irritation of the nasal mucosa
Transdermal testosterone gels	- Less patient resistance - Self-management is possible	- Risk of secondary exposure through skin contact - Skin irritation
Transdermal testosterone ointments	- Less patient resistance - Self-management is possible	- Risk of secondary exposure through skin contact - Skin irritation
Oral testosterone tablets	- Less patient resistance - Self-management is possible - Easy to carry and store	- Systemic side effects (liver dysfunction, polycythemia, hypertension, acne) - Significant first-pass metabolism
Intramuscular injections	- Maintain stable blood testosterone levels - Improved adherence due to infrequent dosing	- Pain or discomfort caused by injections - Systemic side effects (polycythemia) - Injection site reactions - Management at medical institutions (burden of hospital visits)
Subcutaneous injections	- Maintain stable blood testosterone levels - Self-management is possible - Less pain and injection site reactions than intramuscular injections	- Requires mastery of self-injection techniques - Pain or discomfort caused by injections - Systemic side effects (polycythemia) - Injection site reactions
Testosterone pellets	- Maintains stable blood levels - Increased adherence and patient satisfaction	- Minor surgical procedure for insertion/removal - A risk of hematomas and infections

**Table 2 biomolecules-15-01521-t002:** Comparison of Key Recommendations from Major Guidelines on Testosterone Replacement Therapy (TRT).

Organization	Diagnostic Thresholds	Monitoring Protocols	Special Populations/Distinctive Features
Endocrine Society	No universally accepted threshold; diagnosis based on clinical symptoms and consistently low morning TT (<264 ng/dL/9.2 nmol/L)	TT, Ht, PSA before and during TRT	- The potential role of TRT in specific secondary hypogonadal conditions: HIV-associated hypogonadism, opioid-induced hypogonadism - Emphasizes clinical context over strict biochemical cut-offs
AUA	TT < 300 ng/dL (10.4 nmol/L)	TT, Ht, PSA before and during TRT	- Strong emphasis on fertility counseling - Alternative therapies: SERMs, hCG, aromatase inhibitors
ISSAM	TT < 350 ng/dL (12.1 nmol/L) Free T < 220–347 pmol/L (63.5–100 pg/mL)	TT, Ht, PSA before and during TRT	- Prefers long-acting injectable and transdermal formulations - Cautions against oral testosterone - Notes AR CAG repeat polymorphism
EMAS	TT < 8 nmol/L (231 ng/dL): high probability TT 8–12 nmol/L (231–345 ng/dL): borderline Free T < 225 pmol/L (65 pg/mL) supports diagnosis	TT, Ht, PSA Bone density assessment included	- Focus on bone health and fracture risk - Explicit inclusion of osteoporosis screening

Abbreviations: Endocrine Society (ES); American Urological Association (AUA); International Society for the Study of the Aging Male (ISSAM); European Menopause and Andropause Society (EMAS); Total Testosterone (TT); Free Testosterone (Free T); Prostate-Specific Antigen (PSA); Hematocrit (Ht); Selective Estrogen Receptor Modulators (SERMs); Human Chorionic Gonadotropin (hCG); Androgen Receptor (AR).

## Data Availability

No new data were created or analyzed in this study.

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
