# Peer review of "Testosterone and Obesity in an Aging Society"

_biomolecules, 2025, doi:10.3390/biom15111521_

Round 1
Reviewer 1 Report (Previous Reviewer 2)
Comments and Suggestions for Authors
This comprehensive review provides a timely and in-depth examination of the bidirectional relationship between testosterone and obesity, with a particular focus on aging males. The authors have done an excellent job synthesizing a large body of literature to address physiological mechanisms, clinical implications, and therapeutic strategies. The topic is highly relevant given global trends in aging and obesity. The manuscript is generally well-structured, clearly written, and supported by appropriate figures and tables. However, several areas require clarification, expansion, or correction to enhance the manuscript’s rigor and impact.
- The abstract is clear but could be more impactful by including one or two key findings or clinical implications.
- The introduction effectively sets the stage, but consider briefly mentioning the gut microbiota and racial differences here to foreshadow later sections.
- In Sections 2.1.3 and 2.2.2, while the role of inflammation is well highlighted, the discussion on NF-κB and TLR4 signaling could be more detailed. Please consider including a schematic or summarizing table to illustrate the key inflammatory pathways affected by testosterone and obesity.
- In Section (2.1.4), this section is promising but underdeveloped. Please given the emerging importance of the gut-testis axis, a more detailed discussion of human studies and potential clinical implications would strengthen this part.
- In Section 4.3.3, the unexpected finding from the TRAVERSE trial regarding fracture risk deserves more nuanced discussion. Potential mechanisms (e.g., increased physical activity, bone turnover markers) should be explored further, even if speculative.
- In Section 4.4, this section is informative but could be improved by including a comparative table summarizing the key recommendations from Endocrine Society, AUA, ISSAM, and EMAS regarding diagnostic thresholds, monitoring, and special populations.
- In Section 4.5, this is an important and underrepresented area. The manuscript would benefit from citing available evidence (even if limited) on racial variations in testosterone levels, SHBG, and AR polymorphisms, rather than solely highlighting the lack of data.
- Some sentences are overly long or repetitive (e.g., page 3, lines 85–88). Please revise for conciseness and clarity. Minor grammatical errors exist (e.g., “TRT has has been reported” on page 7, line 231).
- Please discuss the potential role of digital health tools or biomarkers for monitoring TRT efficacy and safety in aging populations.
Author Response
Please see the attached file.

Reviewer 2 Report (Previous Reviewer 3)
Comments and Suggestions for Authors
Dear authors, after rereading your manuscript, I noticed that several substantial changes have been made. I have no doubt that your text has significantly improved.
After reading the authors' response and the revised manuscript, my suggestion is that the article is ready for publication.
Kind regards.
Comments on the Quality of English LanguageEnglish must be reviewed.
Author Response
Thank you very much for your kind and encouraging comments. We are pleased to hear that the revised manuscript meets your expectations.
Reviewer 3 Report (Previous Reviewer 4)
Comments and Suggestions for Authors
The authors have adequately addressed all the issues raised.
Author Response
Thank you for confirming that all concerns have been adequately addressed. We appreciate your support and constructive feedback throughout the review process.
Round 2
Reviewer 1 Report (Previous Reviewer 2)
Comments and Suggestions for Authors
1. While generally clear, the manuscript would benefit from a thorough proofread by a native English speaker or professional editing service to correct minor grammatical inconsistencies and improve sentence flow (e.g., "orbitectomized" should be "orchiectomized," "hyposgondalism" typos).
2. In the conclusions, consider adding a sentence on the most pressing future research directions (e.g., long-term TRT trials in obese men, interventional studies on the gut microbiome, integration of genetic data into clinical practice).
3. Section 2.1.4: This is a novel and important topic, but the discussion remains largely descriptive. The manuscript would be significantly strengthened by elaborating on the potential mechanisms of the “gut-testis axis.” How exactly do SCFAs or specific bacterial metabolites influence Leydig cell function or the HPG axis? Discussing the translational potential and challenges. While prebiotics/probiotics are mentioned, what is the current quality of human evidence? Acknowledging the gap between promising animal studies and the need for robust human trials would provide a more balanced perspective.
Author Response
Please see the attached file.

This manuscript is a resubmission of an earlier submission. The following is a list of the peer review reports and author responses from that submission.
Round 1
Reviewer 1 Report
Comments and Suggestions for Authors
Overall rating
The manuscript presents an extensive and well-structured review of the interaction between testosterone, obesity, and aging, addressing physiological mechanisms, clinical consequences, and therapeutic options, with an emphasis on testosterone replacement therapy. The breadth of the topic and the inclusion of recent studies (including the TRAVERSE study) are notable strengths. The language is clear, and the organization facilitates reading, with a logical sequence from pathophysiology to therapeutic implications and future lines of research.
However, to meet the journal's standards, certain methodological and critical aspects would need to be strengthened, avoiding the mere compilation of information.
Strengths
- Broad and up-to-date topic coverage: Multiple mechanisms (inflammation, leptin, SHBG, gut microbiota) are reviewed, and recent evidence from relevant clinical trials is included.
- Clear organization: The structure makes it easy for the reader to follow along from basic concepts to clinical implications.
- Integration of figures and tables: The use of mechanistic schematics and summaries of TRT formulations enhances understanding.
- Bidirectional approach: It describes how obesity decreases testosterone and how low testosterone promotes obesity, highlighting the vicious cycle.
Weaknesses and Recommendations
- Insufficient Critical Depth
- The text summarizes numerous studies but lacks a comparative critical analysis of methodological quality, limitations, or discrepancies between results.
- Recommendation: Incorporate an assessment of study heterogeneity, design, population, and bias, especially in controversial areas such as the cardiovascular and prostate effects of TRT.
- Lack of Inclusion/Exclusion Criteria in the Literature Selection
- The literature search method and criteria for including studies are not explained. This reduces transparency and reproducibility.
- Recommendation: Add a brief section on review methodology, indicating databases, keywords, search period, and selection criteria.
- Insufficient Attention to Subpopulation Differences
- Although cases such as young vs. older obese men are mentioned, variations by ethnicity, comorbidities, or lifestyle are not explored.
- Recommendation: Discuss how the response to TRT and the testosterone-obesity relationship may differ in populations with different metabolic contexts.
- Limited discussion of knowledge gaps and research priorities
- The "Future Research Topics" section is relevant but could be developed with specific hypotheses and potential study designs.
- Recommendation: Propose translational research approaches and randomized clinical trials with clear objectives and intermediate endpoints.
- References
- Although numerous, some sections cite older work when recent meta-analyses exist that could strengthen the evidence.
- Recommendation: Update key citations on molecular mechanisms and the impact of TRT on cardiovascular events and mortality.

Reviewer 2 Report
Comments and Suggestions for Authors
This comprehensive review provides a timely and relevant synthesis of the complex bidirectional relationship between testosterone (T) and obesity, with a specific focus on implications for an aging population. The topic is clinically significant given global demographic shifts and rising obesity rates. The manuscript is generally well-structured, logically organized, and covers key physiological mechanisms, clinical consequences, and therapeutic considerations (particularly TRT). While scientifically sound overall, several areas require clarification, expansion, and improved balance to strengthen the manuscript for publication.
- In Section 2.1.4 (Gut Microbiome), the sentence "However, some clinical research has given different views[96]" is vague. Please briefly specify the nature of the differing views (e.g., conflicting findings on specific microbial changes, or weaker associations in humans compared to mice). Reference [96] (Trigunaite et al. 2015) seems misplaced here, as it is about immune suppression, not microbiome, please ensure the correct citation is used.
- In Section 2.2.4, the section accurately describes the traditional view of increased aromatization in obesity raising E2 and suppressing the HPG axis. It also correctly cites studies showing no significant E2 elevation in some obese hypogonadal men and no change with weight loss. However, the concluding statement (“This raises questions about the simple causal relationship between obesity and increased estradiol levels”) is valid but risks downplaying a potentially important mechanism in some individuals. Furthermore, the sentence “As mentioned earlier, suppression of the HPT system can be mediated through leptin resistance and inflammatory cytokines, a process that can be exacerbated by obesity” seems disconnected from the E2 discussion. Please maintain the nuanced view that aromatase activity and E2 levels can be variable in obese men. Clearly state that while hyperestrogenemia may contribute to HPG suppression in some obese men (explaining why some guidelines consider trial of aromatase inhibitors in select cases), leptin resistance and inflammation are likely more dominant and consistent mechanisms. Please ensure the text flow clearly links back to the primary mechanisms discussed in 2.2.1 and 2.2.2.
- In Section 3.3, the concluding paragraph of Section 3.3 provides practical recommendations but could be clearer and more nuanced:
“TRT may be effective in age-related primary hypogonadism with elevated LH levels.” This is generally accepted, please specify “symptomatic” primary hypogonadism.
“TRT should also be considered in cases of secondary hypogonadism in the aging men, who are not accompanied by an increase in LH and are not obese and who do not use drugs that lower testosterone.” This is too broad. Secondary hypogonadism in aging requires careful investigation to exclude correctable causes (e.g., significant illness, medications, pituitary pathology). TRT is indicated for symptomatic secondary hypogonadism after addressing reversible causes, regardless of obesity status, provided the benefits outweigh the risks. Obesity itself is a common cause of secondary hypogonadism, and weight loss remains first-line. The recommendation here seems to exclude obese men with secondary hypogonadism from TRT consideration, which contradicts standard practice – the focus should be on symptoms, confirmed low T, and addressing obesity first, not an absolute exclusion, so please reframe recommendations emphasizing symptom burden and confirmed biochemical hypogonadism as primary drivers for TRT consideration in all older men (primary or secondary), alongside rigorous evaluation for reversible causes (especially in secondary) and strong encouragement of lifestyle interventions (especially if obese).
Please avoid absolute statements excluding obese men; instead, stress that obesity should be managed concurrently and that TRT efficacy for symptoms may be less predictable in this group if obesity is the primary driver.
- In Sections 3.3 & 4, the text (Section 3.3) appropriately emphasizes weight loss as the first-line intervention for obese men with low T, citing evidence for its efficacy in restoring endogenous T. However, the subsequent statement citing studies advocating TRT for obesity improvement and CV risk reduction (“However, some recent studies have advocated differently...”) needs significant contextualization and balance.
- In Section 4.3.3 (Other TRT Side Effects - Fractures), the discussion of the unexpected TRAVERSE fracture finding is appropriate. The proposed mechanism (increased activity) is plausible. Please consider briefly mentioning other potential speculative mechanisms discussed in the literature (e.g., alterations in bone quality/turnover not captured by BMD, or chance finding).
- The manuscript currently underrepresents the strong consensus from major endocrine societies (e.g., Endocrine Society, ESE) that TRT is not indicated as an obesity treatment. The primary goal of TRT is symptom relief in men with confirmed hypogonadism, not weight loss per se. While TRT may lead to some fat loss and muscle gain in hypogonadal obese men, the effect is often modest and not comparable to dedicated weight loss interventions. Please clearly state that TRT is not approved or recommended as a treatment for obesity itself. Emphasize that its use in obese hypogonadal men should be reserved for those with confirmed hypogonadism (symptoms + consistently low T levels) after careful evaluation and discussion of risks/benefits, and alongside strong encouragement of lifestyle modification for weight loss. Acknowledge the ongoing debate regarding the magnitude of metabolic benefits and the need for more long-term RCTs specifically in obese populations.
- Figure 3 (Age-induced decline): the schematic is helpful but has minor errors/omissions: “Leyding cells” should be “Leydig cells”. “ISIAR + TSPO” is unclear, and it should be “StAR + TSPO”, “Kiss Pin Receptor” should be “Kiss1R” (for consistency with Fig 2 & text).
“Gonadotropins” label after GnRH is ambiguous. Please specify GnRH acts on pituitary to release LH/FSH. Please also consider adding pituitary gonadotrophs. The “Dysfunction of mitochondria” box lacks connection to the specific enzymes mentioned in the text (CYP11A1, etc.).
- For Table 1 (TRT formulations), the information is useful. Improve consistency: “Intranasal gels” Advantages: “Rapid absorption” might be slightly misleading compared to injectables;
“Avoids first-pass” is a key advantage missing here but mentioned in the text.
“Transdermal gels/ointments”: Disadvantages: “Possible migration” should be “Risk of transference/secondary exposure”.
“Oral tablets”: Disadvantages: Add “Significant first-pass metabolism”. “Testosterone pellets”: Disadvantages: “Surgical insertion" should be “Minor surgical procedure for insertion/removal”;
“A risk of hematomas and infections” should be “Risk of infection, extrusion, hematoma”.
Please consider adding a column for “Dosing Frequency” for quick reference.
- Please ensure all abbreviations used in figures are defined in the figure legend or the main text abbreviation list. (e.g., ROS in Fig 2 legend is undefined). While extensive, some references (especially in TRT safety - Sections 4.3.1, 4.3.2) could be updated to include the most recent large-scale studies and meta-analyses (post-2023 where possible).
- Please double-check citations for accuracy (e.g., some placeholder formatting like [96] appears in the gut microbiome section).
- The manuscript is generally well-written. Some sentences are very long and complex (e.g., the long sentence in Section 4.1 starting “Currently, various types...”). Consider breaking these down for clarity. Ensure consistent terminology (e.g., “HPG axis” vs. “HPT system” in 2.2.4).
Reviewer 3 Report
Comments and Suggestions for Authors
Dear Authors,
I have read your manuscript, “Testosterone and Obesity in an Aging Society”, with great attention and genuine interest. First of all, I would like to congratulate you on your initiative. The review tackles a timely and highly relevant topic at the intersection of hormonal regulation, obesity, and aging. Your work brings an important contribution to the field, particularly by highlighting the relationship between testosterone imbalance and key pathological events, such as obesity development.
That said, although the manuscript is thoughtfully elaborated and conceptually solid, there are several aspects that require improvement and deeper discussion. Addressing these issues will not only improve the clarity and impact of the paper but will also help ensure your findings are effectively communicated and well-contextualized within the broader scientific literature. The suggestions below are offered in that spirit.
- The manuscript would benefit from a careful revision of the English. There are several sentences and expressions that sound awkward or are grammatically incorrect. A few examples:
“In clinical meta-analyses, as well as in basic studies, have shown that testosterone deficiency…”. Consider rephrasing for clarity.
“We believe that testosterone may suppresses…”; This should read: “may suppress”.
“This result that the conversion of cholesterol…” What about restructuring this sentence?
“In aging men, it has been reported that a decrease in responsiveness to LH stimulation resulting in the steroidogenic pathway of Leydig cells is impaired.”
I suggest something like this: “In aging men, a reduced responsiveness to LH stimulation has been reported to impair the steroidogenic pathway in Leydig cells.”
“…due to a reduction in the number and function of GnRH neurons, degeneration of kisspeptin neurons in aging men.” Unclear coordination of elements, thus please revise for coherence.
“Obesity and comorbidities are involved in these mechanisms, further complicate this condition.” Come on guys, you can do better!
“...which is mainly caused by the dysfunction function of the testes…”. Dysfunction function? Clearly a typo.
“If these men are obese, the improving obesity may improve…” Unclear and redundant.
“However, there is a risk of secondary exposure due to contact with others,…” “Others” is too vague, don’t you think? Please clarify.
“…during treatment. And it is more patient-acceptable,…” This seems like a typo or awkward phrasing.
“…in hypogonadism men with pre-existing…” Consider using “men with hypogonadism” instead, which, I believe is more idiomatic.
- Figure legends must be placed below the illustrations, as per standard formatting guidelines.
Other few comments and suggestions:
In the Introduction, you state: “Obesity is recognized as a serious health issue in modern society[1–3]. These effects are not limited to changes in appearance…”. Please clarify what “these effects” refer to. Are you speaking of obesity-related metabolic changes? Comorbidities?
- “Testosterone stimulates physical activity through the hypothalamus…” Although I may be able to understand what you want to say, consider being more specific here. Testosterone itself does not directly stimulate physical activity; perhaps rephrase to emphasize its regulatory role in behavior via hypothalamic pathways.
- “Additionally, testosterone has also been shown to inhibit the differentiation of preadipocytes by differentiating macrophages into M2 type macrophages via inhibitory regulatory G protein (Gαi) rather than AR.” Please, provide more details about this relationship, because this is conceptually dense and not easy to follow. Consider unpacking this statement, perhaps in two sentences, and providing a brief explanation of the mechanism.
- There is an overuse of vague reporting phrases such as: “has been reported,” “have suggested,” “has been observed,” “has been confirmed,” etc. Where possible, be more assertive, especially when the cited literature is strong. Academic writing often benefits from confident but evidence-based statements.
- “Based on these things, previous reviews have explicitly…”. “Based on these things” does not seem quite academic and in my opinion it cannot be in your interesting review. What do you think?
Dear authors, your manuscript addresses an important topic and deserves space in the specialized literature. However, substantial revisions are needed, especially concerning academic writing and language clarity. In my opinion, consulting an academic English editor or professional writer could significantly enhance the quality and readability of the review. Please do not be discouraged by these suggestions as they are intended solely to help you strengthen your work. I look forward to reading the revised version and sincerely hope to see it published soon.
Best luck!
Comments on the Quality of English LanguageEnglish must be reviewed.
Reviewer 4 Report
Comments and Suggestions for Authors
Thank you for inviting me to review the manuscript titled „Testosterone and Obesity in an Aging Society”. This manuscript presents a comprehensive and insightful review of the complex relationship between testosterone and obesity. The topic is particularly timely as testosterone has gained significant attention in recent years, with studies exploring its connections to various diseases. The manuscript highlights how obesity affects testosterone production and metabolism, emphasizing the intricate interactions that are still being investigated. Given the rising global interest in this area, the review provides valuable insights into current treatments and offers a solid foundation for future research in this underexplored field.
The manuscript is well-written and well-structured. I would like to offer a few suggestions that could further strengthen the paper and enhance its practical value.
- As the authors have noted, the decline in testosterone is strongly associated with aging. Literature estimates highlight the dynamic nature of this decline. The European Male Aging Study found that total testosterone levels decrease by 0.4% per year, while free testosterone levels drop by 1.3% annually. Additionally, men with obesity experienced a significantly greater reduction in free testosterone levels (by 5.09 nmol/L; p < 0.001) compared to those with normal weight. (Wu FC et al. J. Clin. Endocrinol. Metab. 2008, 93, 2737–2745).
- Following the publication of the TRAVERSE study results (Lincoff et al.), the cardiovascular risk is no longer considered as controversial as it once was. Therefore, I believe the information regarding the magnitude of this risk should be revised to reflect this. Additionally, Lincoff's study appears twice in the reference list, as entries 38 and 221, and this duplication should be corrected.
- There is a lack of a summary regarding guidelines for testosterone replacement therapy (TRT) based on testosterone levels. Could you consider adding a paragraph on this topic? For example, Hackett (BSS), Dean (ISSM), Salonia (EAU guidelines), and Kanakis (EMAS position statement) all provide slightly different insights, leading to some controversy regarding when to initiate treatment, which preparations to use first, and how to monitor therapy.
- I suggest adding a clinical definition of hypogonadism, along with the diagnostic criteria. The term "late-onset hypogonadism," which links reduced testosterone levels with aging, is not mentioned in the paper. A useful reference for the three groups of hypogonadism symptoms can be found in Wu FC's article (N Engl J Med 2010; 363:123-35).
- The relationship between testosterone and lipid metabolism is not addressed. In men with obesity, low testosterone levels can lead to increased body fat accumulation. Additionally, both obesity and aging are factors that contribute to sexual and erectile dysfunction, which are significant clinical symptoms of low testosterone. This often prompts many men to seek TRT (Biernikiewicz M et al. J Clin Med. 2024 3;13(7):2087).